# Low-Temperature Regulates the Cell Structure and Chlorophyll in Addition to Cellulose Metabolism of Postharvest Red *Toona sinensis* Buds across Different Seasons

**DOI:** 10.3390/ijms25147719

**Published:** 2024-07-14

**Authors:** Qian Zhao, Fu Wang, Yifei Wang, Xiulai Zhong, Shunhua Zhu, Xinqi Zhang, Shuyao Li, Xiujuan Lei, Zhenyuan Zang, Guofei Tan, Jian Zhang

**Affiliations:** 1Faculty of Agronomy, Jilin Agricultural University, Changchun 130118, China; zhaoqian@mails.jlau.edu.cn (Q.Z.); wangfu@mails.jlau.edu.cn (F.W.);; 2Institute of Horticulture, Guizhou Provincial Academy of Agricultural Sciences, Guiyang 550006, China; 3College of Chinese Medicinal Materials, National and Local Joint Engineering Research Center for Ginseng Breeding and Development, Jilin Agricultural University, Changchun 130118, China; 4Department of Biology, University of British Columbia, Okanagan, Kelowna, BC V1V 1V7, Canada

**Keywords:** *Toona sinensis*, refrigeration, paraffin sections, greening, fibrosis, gene expression

## Abstract

Postharvest fibrosis and greening of *Toona sinensis* buds significantly affect their quality during storage. This study aimed to clarify the effects of low-temperature storage on postharvest red TSB quality harvested in different seasons. Red TSB samples were collected from Guizhou province, China, 21 days after the beginning of spring (Lichun), summer (Lixia), and autumn (Liqiu), and stored at 4 °C in dark conditions. We compared and analyzed the appearance, microstructure, chlorophyll and cellulose content, and expression levels of related genes across different seasons. The results indicated that TSB harvested in spring had a bright, purple-red color, whereas those harvested in summer and autumn were green. All samples lost water and darkened after 1 day of storage. Severe greening occurred in spring-harvested TSB within 3 days, a phenomenon not observed in summer and autumn samples. Microstructural analysis revealed that the cells in the palisade and spongy tissues of spring and autumn TSB settled closely during storage, while summer TSB cells remained loosely aligned. Xylem cells were smallest in spring-harvested TSB and largest in autumn. Prolonged storage led to thickening of the secondary cell walls and pith cell autolysis in the petioles, enlarging the cavity area. Chlorophyll content was higher in leaves than in petioles, while cellulose content was lower in petioles across all seasons. Both chlorophyll and cellulose content increased with storage time. Gene expression analysis showed season-dependent variations and significant increases in the expression of over half of the chlorophyll-related and cellulose-related genes during refrigeration, correlating with the observed changes in chlorophyll and cellulose content. This research provides valuable insights for improving postharvest storage and freshness preservation strategies for red TSB across different seasons.

## 1. Introduction

*Toona sinensis* (A. Juss.) Roem., a perennial deciduous tree belonging to the Meliaceae family, is widely distributed in Southeast Asia and has been cultivated in China for over 2000 years [1,2]. The initial buds of *T. sinensis,* referred to as TSB, initially exhibit red or purple which later turn green as they mature [3]. Based on their color, TSB buds are divided into two distinct groups: the red TSB and the green TSB [4,5]. Among these, red TSB is notable for its bright color, rich flavor, crispiness, and juiciness. The entire young shoots, tender leaflets, and petioles of TSB are all edible, making it a favored woody vegetable among the Chinese [6,7,8,9]. In addition, TSB is rich in protein, amino acids, vitamins, anthocyanins, chlorophyll, dietary fiber and flavonoids, terpenes, phenols, and other pharmacologically active compounds [10,11]. These compounds confer various medicinal benefits to TSB, such as anti-oxidation [12], anti-inflammatory [13], anti-blood glucose [14], anti-cancer [15], and anti-tumor [16]. Thus, TSB is widely regarded as a medicinal and edible woody vegetable.

TSB is harvested over an approximately three-week period and often picked around the Qingming Festival, which is at the beginning of April. Advances in greenhouse technology and global warming have enabled year-round cultivation of TSB [17]. Nevertheless, the effective postharvest management of TSB remains a significant challenge. The fresh TSB, due to its high water content and vigorous metabolic activity, is perishable, with a shelf life of only 1–2 days at room temperature. Common postharvest issues such as fibrosis, decay, browning, and greening seriously affect the quality and marketability of TSB [18,19]. Chlorophyll and cellulose metabolism are likely contributors to these postharvest issues, affecting the color, texture, and taste of TSB [20,21].

Low-temperature storage is the most commonly used preservation method for fruit and vegetables, as it is both economical and convenient. This method can prevent tissue browning, maintain quality, slow down the aging process, and prolong the storage period and shelf life of fruits and vegetables [22]. Ning et al. found that low-temperature refrigeration not only maintained the firmness of Hami melon, but also inhibited the production of malondialdehyde and H_2_O_2_, thereby significantly reducing its decay rate, weight loss, and relative conductivity, further extending the shelf life of Hami melon [23]. Hong et al. studied the effects of cold storage on postharvest pears, revealing that it mitigated fruit decay and extended shelf life [24]. Xiao et al. found that cold storage can effectively prolong the shelf life of fresh sweet corn [25]. Despite numerous studies on the postharvest storage of TSB, little research has focused on seasonal variations in physiological indices of red TSB, particularly in terms of cellular biology, physiology, biochemistry, and gene expression. At the same time, there are gaps in the understanding of chlorophyll and cellulose metabolism in postharvest TSB.

Therefore, this study utilized red TSB harvested from Guizhou province during Lichun, Lixia, and Liqiu, storing them in dark conditions at 4 °C. The study observed changes in color, chlorophyll and cellulose contents, and microstructure with the leaflet and petiole of red TSB during refrigeration across different seasons. At the same time, according to our group’s previously established transcriptome database of TSB [3], the chlorophyll-related and cellulose-related genes in TSB were retrieved and identified, and these gene expression levels were further analyzed in postharvest TSB during refrigeration across different seasons. The aim was to provide a theoretical foundation for optimizing postharvest storage and quality control of TSB across different seasons, ensuring its optimal preservation throughout different seasons.

## 2. Results

### 2.1. Blast of Chlorophyll-Related and Cellulose-Related Genes

According to the TSB transcriptome database, 14 chlorophyll-related genes (*TsALAD*, *TsHEMA*, *TsPBGD*, *TsUROD*, *TsUROS*, *TsCPOX*, *TsCHLH*, *TsCHLI*, *TsCHLD*, *TsChIM*, *TsDVR*, *TsGGR*, *TsCHLG*, *TsCAO*) and 14 cellulose-related genes (*TsCesA1*, *TsCesA2*, *TsCesA3*, *TsCesA4*, *TsCesA, TsCesA6*, *TsCesA7*, *TsCesA8*, *TsCslB4*, *TsCslD3*, *TsCslD5*, *TsCslE6*, *TsCslG2*, *TsCslG3*) were identified, as detailed in Appendix A. These genes were translated into amino acid sequences and compared with those of other woody plants on the NCBI online website. The results showed high consistency (80.31% and 96.32%) with woody plant sequences, except for *TsCslE6* and *TsCslG2* genes.

### 2.2. Color Changes of Red TSB during Postharvest Storage

In terms of color and appearance, spring red TSB has purplish-red leaflets and petioles, while summer and autumn red TSBs only have purple leaflets and green petioles. During postharvest storage, red TSB leaflets wilt, causing color fading within 1 day. Spring red TSB notably changes color from purple-red to green after 1 day of storage, becoming more evident after 3 days, unlike the summer and autumn red TSBs. The purple-red of the leaflets in the summer red TSB is between the spring and autumn red TSB. The petioles of the summer TSB are light green, while the autumn red TSBs’ petioles are almost close to dark green. During postharvest storage, the leaflets of red TSB display phenomena like wilting, resulting in a fading color within just 1 day. Notably, the spring red TSB undergoes a significant transformation, the color turned from purple-red to green when stored for 1 day, and the green was especially obvious when stored for 3 days. Interestingly, there was no obvious greening phenomenon in the summer and autumn red TSBs during storage (Figure 1).

### 2.3. Structural Characteristics of Red TSB during Postharvest Storage

Leaflet parenchyma cells in TSB, rich in chloroplasts, are known as chlorenchyma and perform photosynthesis. Spring red TSB’s parenchyma cells are loosely arranged when stored for 0 days, contrasting with the tighter arrangement in summer and autumn TSBs. During storage, the spring TSB’s parenchyma cells enlarge and compact, while summer TSBs show increased intercellular space leading to a looser arrangement after 1 day of storage and tightening after 3 days of storage. In autumn, red TSB’s parenchyma cells increase and tightly arrange (Figure 2).

Cellulose is the main structural component of plant cell walls, providing integrity. Safranine-fast green staining not only colors the cell wall blue-green but also reveals cellulose distribution and accumulation. Paraffin section results showed cellulose mainly in the xylem of the petiole, with less blue-green xylem in spring red TSB than in summer and autumn red TSB. Intriguingly, spring red TSB’s petiole xylem cells are the smallest, while autumn red TSB’s are the largest. Over time, red TSB’s petiole xylem cells enlarge, and secondary cell walls thicken (Figure 3A). This finding highlights the senescence of TSB during storage.

Notable changes in pith cells of red TSB can be observed. Pith cells in spring and autumn red TSB are closely arranged, contrasting with the loose arrangement in summer TSB after 0 days of storage. During postharvest storage, pith cells show significant changes. The petiole in the spring red TSB has more closely arranged pith cells that increase in number. Summer red TSB exhibits pith cell autolysis, forming large hollows that grow over time. Autumn red TSB’s pith cells enlarge and thicken cell walls after 1 day of storage, showing signs of cellular autolysis (Figure 3B).

### 2.4. Chlorophyll and Cellulose Content of Red TSB during Postharvest Storage

Chlorophyll, crucial for photosynthesis, is indicative of plant health. Chlorophyll a and b content indicates photosynthesis quality and postharvest red TSB evaluation. Variations in chlorophyll content were observed during the refrigeration of postharvest red TSB across different seasons, which is essential for assessing its quality. In this study, we measured the chlorophyll content of postharvest red TSB during refrigeration across different seasons, noting significant variations.

Chlorophyll content in leaflets consistently exceeds that in petioles across different harvesting periods. Chlorophyll a content significantly surpassed chlorophyll b, serving as the primary pigment in photosynthesis. It efficiently absorbs light energy and converts it into chemical energy, crucial for plant growth. Hence, higher chlorophyll a content is vital for efficient light energy conversion. When stored for 0 days, chlorophyll a and chlorophyll b content in leaflets of summer red TSB exceeded that of spring and autumn red TSB, with the spring red TSB showing the lowest levels. Chlorophyll a and b contents in spring, summer, and autumn were 0.48 and 0.18 mg·g^−1^, 1.21 and 0.47 mg·g^−1^, 0.86 and 0.35 mg·g^−1^, respectively. Chlorophyll content in postharvest red TSB exhibited seasonal variations with the extension of storage time. After 3 days of storage, chlorophyll a content increased in leaflets of red TSB (excluding spring TSB), and chlorophyll b content also rose, though the difference was not significant (Figure 4). After 3 days of storage, chlorophyll content was highest in the summer and lowest in the spring for red TSB. Furthermore, chlorophyll content in petioles of red TSB remained relatively stable across different seasons.

To study cellulose accumulation in postharvest red TSB across different seasons, we measured cellulose content in leaflets and petioles during refrigeration. The results revealed varying cellulose patterns, highest in summer and lowest in spring. During postharvest storage, cellulose content slightly decreased in spring and summer but increased with other treatments. Notably, the cellulose content of red TSB significantly increased in summer and autumn by 39.70% and 5.40%, respectively. Additionally, petioles consistently contained higher cellulose than leaflets. At 0 days of storage, petioles had higher cellulose content than leaflets by 49.64, 47.63, and 52.87 mg·g^−1^ in spring, summer, and autumn, respectively. After 3 days of storage, this difference widened further, with petioles maintaining higher cellulose content than leaflets by 57.62, 129.99, and 60.80 mg·g^−1^, respectively (Figure 4).

### 2.5. Chlorophyll-Related Genes Expression of Red TSB during Postharvest Storage

From the transcriptome databases of red TSB, we selected 14 genes involved in chlorophyll synthesis and degradation, including *TsALAD*, *TsHEMA*, *TsPBGD*, *TsUROD*, *TsUROS*, *TsCPOX*, *TsCHLH*, *TsCHLI*, *TsCHLD*, *TsChIM*, *TsDVR*, *TsGGR*, *TsCHLG,* and *TsCAO*. The results indicated significant variation in expression levels of chlorophyll-related genes during postharvest storage across different seasons and tissues.

During postharvest storage in spring red TSB, the expression levels of chlorophyll synthesis-related genes *TsHEMA*, *TsUROS*, and *TsCAO* significantly increased in leaflets and petioles. Conversely, the expression level of the chlorophyll degradation-related gene *TsChIM* significantly decreased in both leaflets and petioles, while the expression level of the *TsCPOX* gene exhibited a substantial increase (Figure 5). During postharvest storage in summer red TSB leaflets, the expression levels of chlorophyll metabolism genes (*TsHEMA*, *TsPBGD*, *TsUROD*, *TsUROS*, *TsCPOX*, *TsCHLH*, and *TsCAO*) showed a nonlinear increasing trend during postharvest storage. In contrast, only the expression levels of chlorophyll degradation genes *TsCPOX* and *TsDVR* showed a nonlinear decrease in the petioles of red TSB (Figure 6). In autumn, the expression levels of all genes in red TSB leaflets, except for *TsHEMA*, *TsCPOX*, *TsChIM*, *TsGGR*, and *TsCAO*, significantly increased during refrigeration. However, the expression levels of the *TsCHLH* gene notably decreased. Similarly, in the petioles of red TSB in autumn, the expression levels of *TsHEMA*, *TsUROD*, *TsCPOX*, *TsCHLG*, and *TsCAO* genes significantly increased, while the expression levels of *TsCHLG* and *TsCAO* genes were notably elevated in red TSB petioles. Overall, the mentioned genes exhibited significant upregulation, whereas the expression levels of *TsPBGD*, *TsCHLD*, and *TsDVR* genes underwent significant downregulation (Figure 7).

### 2.6. Cellulose-Related Genes Expression in Red TSB

Based on the transcriptome database of red TSB, a series of cellulose synthesis genes were obtained, including *TsCesA1*, *TsCesA2*, *TsCesA3*, *TsCesA4*, *TsCesA5*, *TsCesA6*, *TsCesA7*, *TsCesA8*, *TsCslB4*, *TsCslD3*, *TsCslD5*, *TsCslE6*, *TsCslG2,* and *TsCslG3*. For further investigation, qPCR was used to detect the expression levels of these genes in the leaflets and petioles of red TSB across different seasons. The results showed significant changes in the expression levels of cellulose-related genes as storage time increased (Figure 6).

The cellulase genes *TsCesA2*, *TsCesA4*, *TsCesA5*, *TsCesA6*, *TsCesA7*, and *TsCesA8*, along with the hemicellulase genes *TsCslD3*, *TsCslE6*, and *TsCslG2*, were highly involved in cellulose synthesis in the petioles of red TSB during the spring postharvest storage process. These genes played a crucial role in cellulose synthesis in the petioles of red TSB after harvesting, with the exception of *TsCesA1*, *TsCesA3*, and *TsCslB4*. Other genes were also found to have a significant impact on cellulose synthesis in the petioles of postharvest red TSB (Figure 8). Besides the hemicellulose gene *TsCslB4*, the rest of the cellulose metabolism genes showed a nonlinear increase in gene expression in the leaflets and petioles of red TSB during the summer season (Figure 9). The expression patterns of *TsCesA1*, *TsCesA3*, *TsCslD5*, *TsCslE6*, and *TsCslG3* genes differed between the leaflets and petioles of red TSB in the autumn season. These genes were significantly upregulated in the leaflets but downregulated in the petioles. The expression levels of *TsCesA2*, *TsCesA4*, *TsCesA5*, and *TsCesA6* genes also differed between the leaflets and petioles of red TSB in the autumn season. The expression levels of *TsCesA7*, *TsCesA8*, *TsCslD3*, and *TsCslG2* genes increased nonlinearly during refrigeration of leaflets and petioles of red TSB in autumn. However, the *TsCslB4* gene exhibited significant down-regulation during postharvest refrigeration of leaflets and petioles of red TSB in the autumn (Figure 10).

## 3. Discussion

The color variation of red TSB throughout the seasons may be influenced by its growth and development stages, as well as the unique climate of Guizhou. In spring, the growth rate of red TSB accelerates notably, resulting in its initial buds appearing red or purple-red [3]. Moreover, the low temperature in spring effectively promotes the synthesis and accumulation of anthocyanins [26]. During summer and autumn, the growth rate of red TSB slows, and it turns green upon maturity [3]. Sufficient light during these seasons promotes chlorophyll accumulation in TSB, resulting in their characteristic green color [27]. Color is a crucial sensory attribute of postharvest fruit and vegetables, crucial for consumers to evaluate maturity, freshness, marketability, and nutritional value. The gradual darkening of TSB during storage may be attributed to the accumulation of phenolics in postharvest red TSB, ultimately leading to this phenomenon [28]. Moreover, the green discoloration observed in postharvest red TSB during refrigeration could be related to the aging process, as it is commonly associated with fruit and vegetable storage [20].

The degree of parenchyma development in the leaflet, a major site of photosynthesis, correlates with the intensity of photosynthesis, respiration, and transpiration processes [29]. Postharvest red TSB wilted after 1 day of storage. In contrast to spring and autumn red TSB, parenchyma cells in summer red TSB exhibited loose arrangement during refrigeration, indicating accelerated water loss due to postharvest respiration and transpiration leading to wilting [30,31]. Additionally, as the plant undergoes lignification during refrigeration, its tissues or organs experience reduced water content due to increased lignification [32]. Thus, water loss led to expanded intercellular spaces, resulting in loose parenchyma arrangement. Moreover, respiration and transpiration rates of summer red TSB during postharvest storage were conspicuously lower compared to spring and autumn red TSB, suggesting a potential difference in shelf life. Cellulose primarily deposited in the xylem led to the gradual thickening of secondary cell walls in the xylem cells of red TSB petioles over storage time. Furthermore, the autolysis phenomenon was observed in the pith cells during refrigeration of summer and autumn TSB petioles. This indicates that during postharvest storage of *T. sinensis* buds, cytoplasmic substances gradually dissolve as cells undergo senescence, leading to void formation. Chlorophyll is crucial for maintaining the storage and preservation of green leafy vegetables. However, low temperatures and dark conditions accelerate chlorophyll decomposition, leading to yellowing [33]. Yet, green discoloration observed in postharvest red TSB during refrigeration is considered a defect affecting its marketability. Interestingly, chlorophyll content in postharvest red TSB during refrigeration showed an increasing trend across different seasons, unlike observations in *Brassica oleracea* L. var. *italica Plenck* [34], *Brassica oleracea* var. *capitata* Linnaeus [35], and *Brassica rapa* subsp. chinensis [36]. This divergence may be attributed to the short-term storage of red TSB, causing a temporary increase in chlorophyll content. This may relate to their unique physiological properties. Extended storage time may lead to chlorophyll decomposition and chlorosis of red TSB due to its inherent instability. Furthermore, prolonged dark storage of green leafy vegetables leads to chlorophyll degradation and reduction in content.

Dietary fiber, a polysaccharide indigestible by the human gastrointestinal tract, is broadly classified into soluble and insoluble types [37], both highly beneficial for human health. Insoluble dietary fiber primarily comprises cellulose, hemicellulose, and lignin, serving as essential components of plant cell walls that support growth and stress resistance. However, the synthesis and accumulation of these components greatly impact the quality of fruit and vegetables [21,38]. Cellulose content showed a gradual increase throughout the postharvest storage period of red TSB. This increase may result from a cross-linking reaction between cellulose and other substances during refrigeration [39]. Similarly, the cellulose content in the stems of *Dendrobium officinale* increased by 1.8 times during postharvest storage compared with that at the beginning of fresh harvest [40], which was consistent with the changing trend of cellulose content in the red TSB observed.

Chlorophyll-related and cellulose-related gene expression levels directly or indirectly regulate chlorophyll and cellulose synthesis and accumulation [41,42]. During refrigeration, expression levels of chlorophyll-related genes were up-regulated in postharvest red TSB, contrasting with mechanisms observed in other postharvest vegetables like tomato and broccoli. Atmospheric cold plasma treatments were employed to inhibit chlorophyll degradation and significantly down-regulate expression levels of chlorophyll metabolism-related genes (*SlCLH1*, *SlPPH*, and *SlRCCR*) in postharvest tomato [43]. Decreased chlorophyll content and down-regulation of chlorophyll-related enzymes and gene expression levels during postharvest storage of broccoli led to yellowing phenomenon [44]. This contrasts with trends in chlorophyll-related gene expression levels in postharvest red TSB during refrigeration, possibly attributable to water loss during storage. Cellulose content and expression levels of cellulose-related genes significantly increased in postharvest red TSB during refrigeration, a trend observed in many horticultural crops during storage [45,46]. Due to temporal and tissue-specific gene expression [47,48], expression levels of chlorophyll-related and cellulose-related genes in postharvest red TSB’s leaflets and petioles across different seasons varied, yet generally aligned with chlorophyll and cellulose content changes. Additionally, normal plant development may influence chlorophyll and cellulose content changes and gene expression levels.

## 4. Materials and Methods

### 4.1. Material Collection and Pretreatment

Red TSB with uniform growth conditions, no pests, no diseases, and no mechanical damage were selected (Red *T. sinensis* plant was cultivated by Guizhou Muye Pastoral Agricultural Science and Technology Development Company in Hongguang Village, Banqiao Town, Zhijin County, Guizhou Province (105.71° E, 26.79° N)). Red TSB was harvested 7 days after the beginning of spring (25 February to 28 February), summer (27 May to 30 May), and autumn (29 August to 1 September), packed in polyethylene bags, and stored in dark conditions at 4 °C. Samples were collected at the same time (7:00 a.m.) on 0 days (day of harvest), 1 day, 2 days, and 3 days of storage, with three biological replicates per sampling point. Postharvest color changes of red TSB were observed across different seasons during refrigeration. Leaflets and petioles from different seasons of postharvest red TSB were fixed in FAA fixative for tissue sectioning to observe structure changes [49]. At the same time, leaflets and petioles were flash-frozen in liquid nitrogen and stored at −80 °C for subsequent total RNA extraction and determination of chlorophyll and cellulose content.

### 4.2. Color and Structure Observation of Red TSB

Color changes in postharvest red TSB across different seasons during refrigeration were observed, followed by preparation of paraffin sections and histochemical staining. Leaflets and petioles from corresponding parts of postharvest red TSB were fixed in FAA fixative for 24 h, dehydrated in ethanol gradient, embedded in paraffin, sectioned at 4 µm thickness using a microtome, stained with safranin-fast green, dehydrated again in ethanol gradient, and finally observed and photographed under an optical microscope (NIKON ECLIPSE E100, Nikon Instruments Shanghai Company, Shanghai, China).

### 4.3. Determination of Chlorophyll and Cellulose Content

Approximately 0.2 g of leaflets (midrib removed) or petioles from postharvest red TSB were weighed and ground into homogenate with 95% ethanol until the tissue turned white. After standing for 5 min, the homogenate was filtered into a 0.025 L beaker, washed with a small amount of 95% ethanol, and topped up to 0.025 L with 95% ethanol. Each sample included three biological replicates. The OD values at 665 nm and 649 nm wavelengths were measured using an ultraviolet spectrophotometer, and chlorophyll content in red TSB was determined with 95% ethanol as the blank control [50,51,52]. Results were reported on a fresh weight basis using the following formulas:$$\rho_{a}=13.95A_{665}-6.88A_{649};$$
$$\rho_{b}=24.96A_{649}-7.32A_{665};$$
$$chlorophyll\;a\;content\left(\mathrm{mg}/g\right)=\frac{\rho_{a}\times v}{1000\times m};$$
$$chlorophyll\;b\;content\left(\mathrm{mg}/g\right)=\frac{\rho_{b}\times v}{1000\times m};$$
chlorophyll tatol content (mg/g)=chlorophyll a content+chlorophyll b content.

Based on the method of Lv et al. [53], the cellulose content of postharvest red TSB was analyzed. A total of 0.3 g red TSB samples were quickly homogenized at room temperature, and crude cell wall substances were obtained through processes including water bath, centrifugation, and washing. These substances were then soaked, precipitated, washed, and dried to obtain cell wall substances (CWM). Finally, 0.005 g of cell wall material was weighed for cellulose content determination in postharvest red TSB and the results were expressed based on dry weight.

### 4.4. Total RNA Extraction and cDNA Synthesis of Red TSB

Postharvest red TSB samples stored at −80 °C were ground into powder using liquid nitrogen in sterilized mortars. Approximately 0.1 g of the ground sample was transferred to a 0.002 L non-enzymatic centrifuge tube for total RNA extraction using a using RNA extraction kit (Beijing Huayueyang Biotechnology Co., Ltd., Beijing, China, GX type). The integrity of total RNA was confirmed by 1.5% agarose gel and the concentration was determined using the Nanodrop ND-100 (Nanodrop Technology Inc., Wilmington, DE, USA). Subsequently, cDNA was synthesized using the HiScript III 1st Strand cDNA Synthesis Kit (+gDNA wiper) (Vazyme Biotech Co., Ltd., Nanjing, China), according to the manufacturer’s protocol.

### 4.5. Gene Identification and Primers Design

Gene sequences related to chlorophyll and cellulose biosynthesis were retrieved from the transcriptome database of red TSB according to their metabolic pathways. The open reading frame and amino acid sequence of these genes were determined using BioXM2.7.1 software, followed by comparison with the NCBI database to verify their accuracy and integrity. Primers for qPCR analysis of chlorophyll-related and cellulose-related genes were designed using Primer Primers 6 software, and the primers (Appendix A) were synthesized by Genscript Biotech Corporation (Nanjing, China).

### 4.6. Expression Analysis of Chlorophyll-Related and Cellulose-Related Genes

Real-time fluorescence quantitative PCR (CFX ConnectTM Real-Time System, Bio-Rad, Hercules, CA, USA) was conducted to detect the expression levels of the chlorophyll-related and cellulose-related genes in postharvest red TSB during refrigeration across different seasons. The ChamQ Universal SYBR qPCR Master Mix Kit (Vazyme, Nanjing, China) was utilized with cDNA as a template and *TsActin* as an internal standard. The *TsActin* gene of red TSB was co-amplified with the target gene. Each sample was set to three biological replicates. The qPCR reaction mixture totaled 20 µL, comprising 2 × SYBR qPCR Master Mix (10 µL), templated cDNA (2 µL), upstream primers (0.5 µL), downstream primers (0.5 µL), ddH_2_O (7 µL). The PCR reaction program was as follows: pre-denaturation at 95 °C for 5 min; denaturation at 95 °C for 10 s, annealing at 54 °C for 30 s, and extension at 65 °C for 10 s with 40 cycles; and the samples were stored at 8 °C. Relative expression level was calculated using the 2^−ΔΔCT^ method, where ΔCT = C*_Ttarget gene_* − C*_TActin_* [54].

### 4.7. Data Analysis

Microsoft Excel 2007 software was used for data processing. IBM Statistics 20.0 software was used to analyze the significant difference between groups, and the significant level was set to *p* < 0.05. Additionally, Origin 2021 software was employed for mapping purposes.

## 5. Conclusions

This study utilized spring, summer, and autumn red TSB samples as experimental materials, harvested for storage at 4 °C in dark conditions. We conducted a comparative analysis of external color, internal anatomical structure, chlorophyll, and cellulose content, as well as the expression level of related genes across different seasons during refrigeration. Seasonal differences in the cellular changes reflect the physiological responses of the red TSB during postharvest storage. The observed increase in chlorophyll and cellulose content during postharvest storage indicates ongoing metabolic processes and structural modifications, potentially influenced by gene expression patterns. These findings contribute to our understanding of postharvest management strategies for red TSB across different seasons. The model diagram of TSB during refrigeration across different seasons is shown in Figure 11.

## Figures and Tables

**Figure 1 ijms-25-07719-f001:**
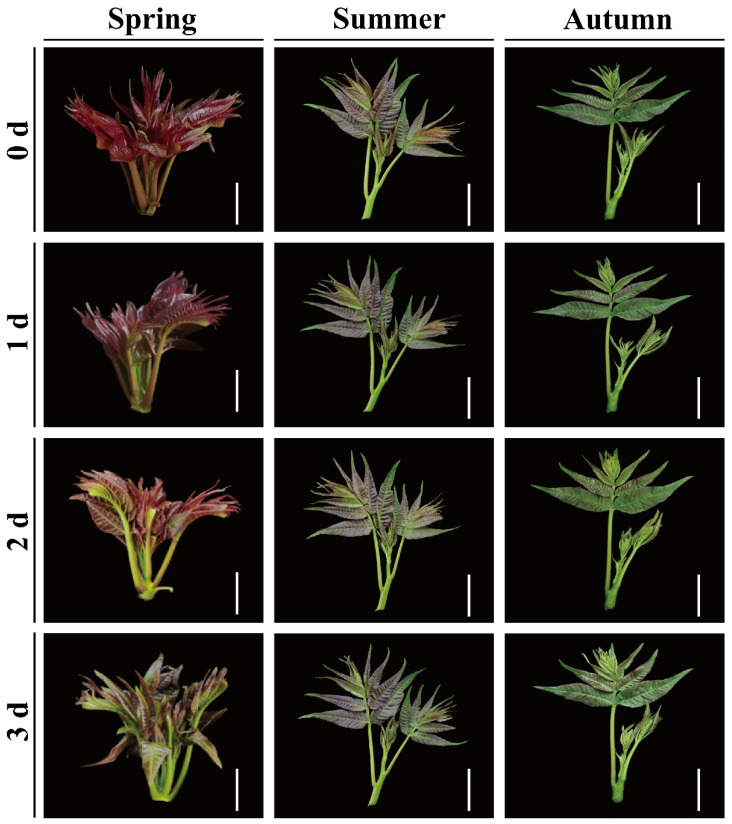
Color change of the buds of red *T. sinensis* harvested at different times of the year, during four days of refrigerated storage (4 °C) in the dark. Bar = 4 cm.

**Figure 2 ijms-25-07719-f002:**
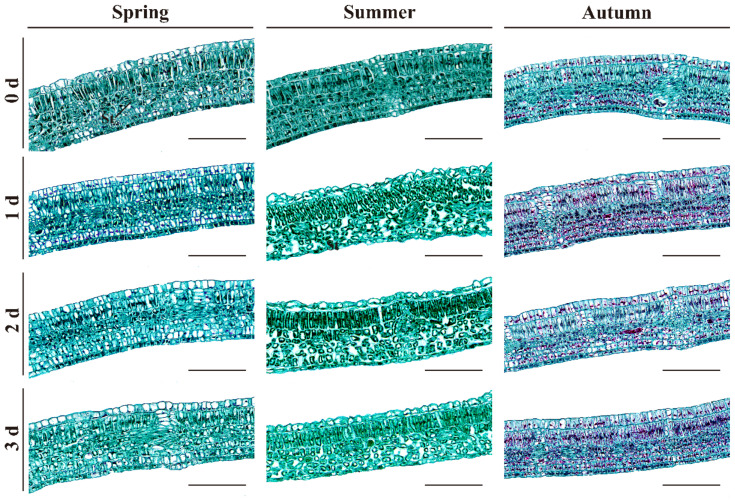
Mesophyll structure of the leaflets from the buds harvested at different times of the year, during four days of refrigerated storage (4 °C) in the dark. Bar = 100 µm.

**Figure 3 ijms-25-07719-f003:**
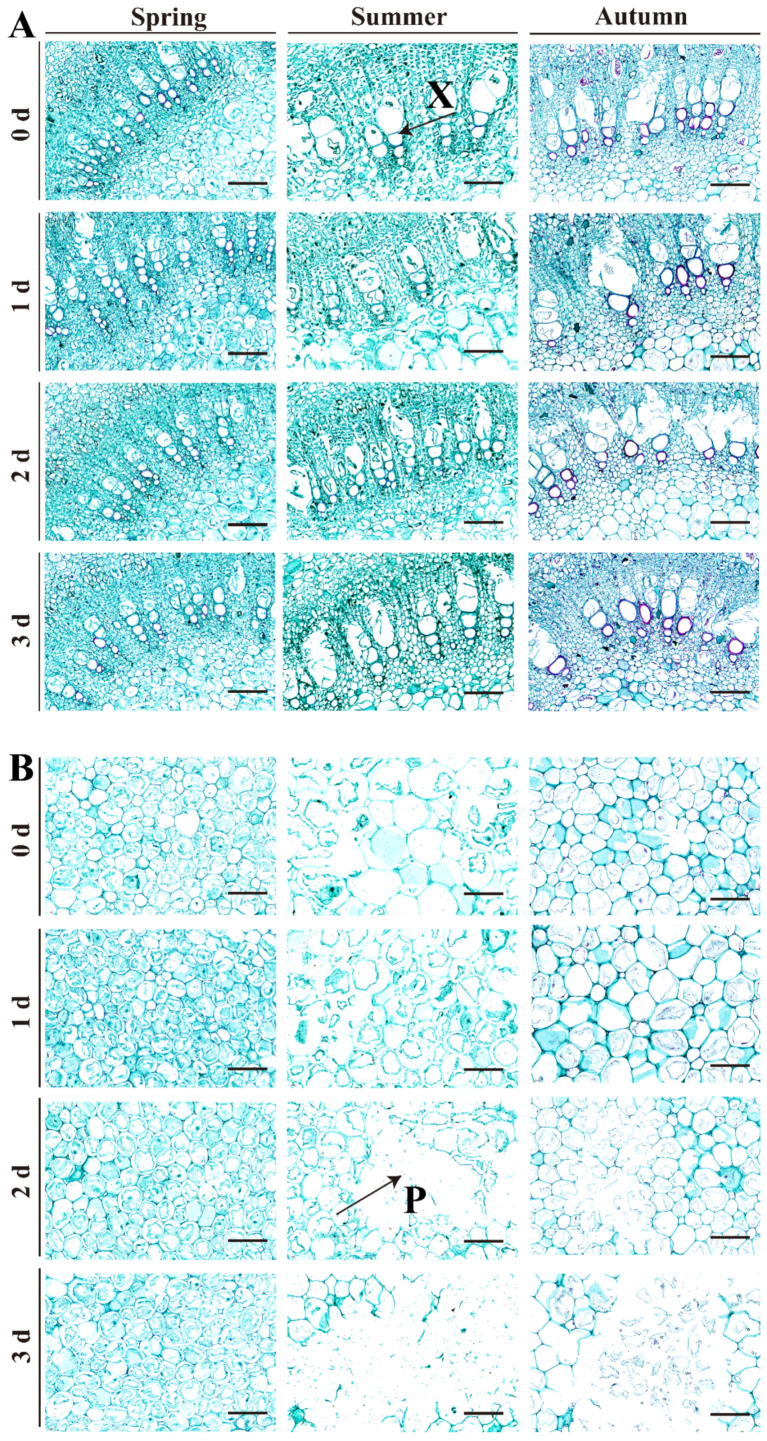
Petiole structure of red TSB harvested at different times of the year, during four days of refrigerated storage (4 °C) in the dark. (**A**) The xylem of petiole in postharvest red TSB during refrigeration across different seasons. (**B**) The pith of petiole in postharvest red TSB during refrigeration across different seasons. Xylem (X) and Pith (P) were marked in the figure. Bar = 100 µm.

**Figure 4 ijms-25-07719-f004:**
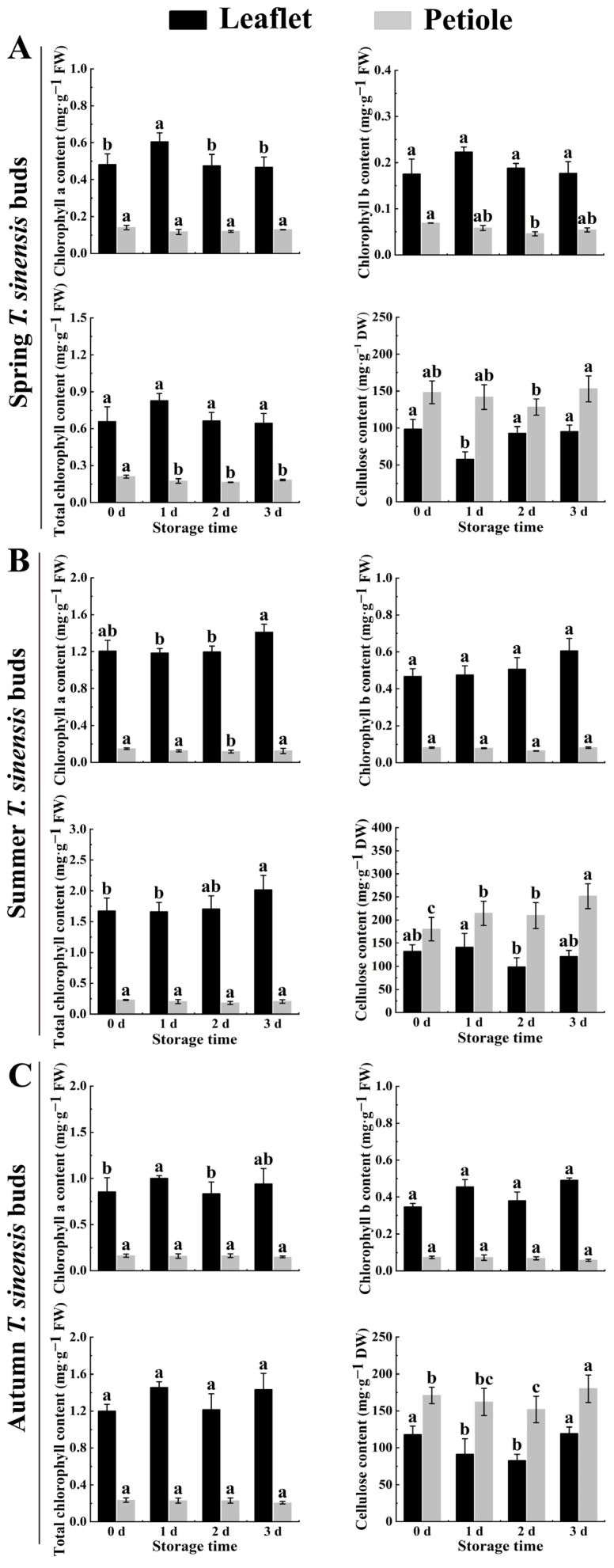
The variation in chlorophyll and cellulose content within the leaflet and petiole of postharvest red TSB during refrigeration across different seasons. The “a”, “b” and “c” indicate significant differences between treatments (*p* < 0.05). The chlorophyll a, chlorophyll b, total chlorophyll and cellulose content during refrigeration which are presented in (**A**), of spring red TSB; (**B**), of summer red TSB and (**C**), of autumn red TSB.

**Figure 5 ijms-25-07719-f005:**
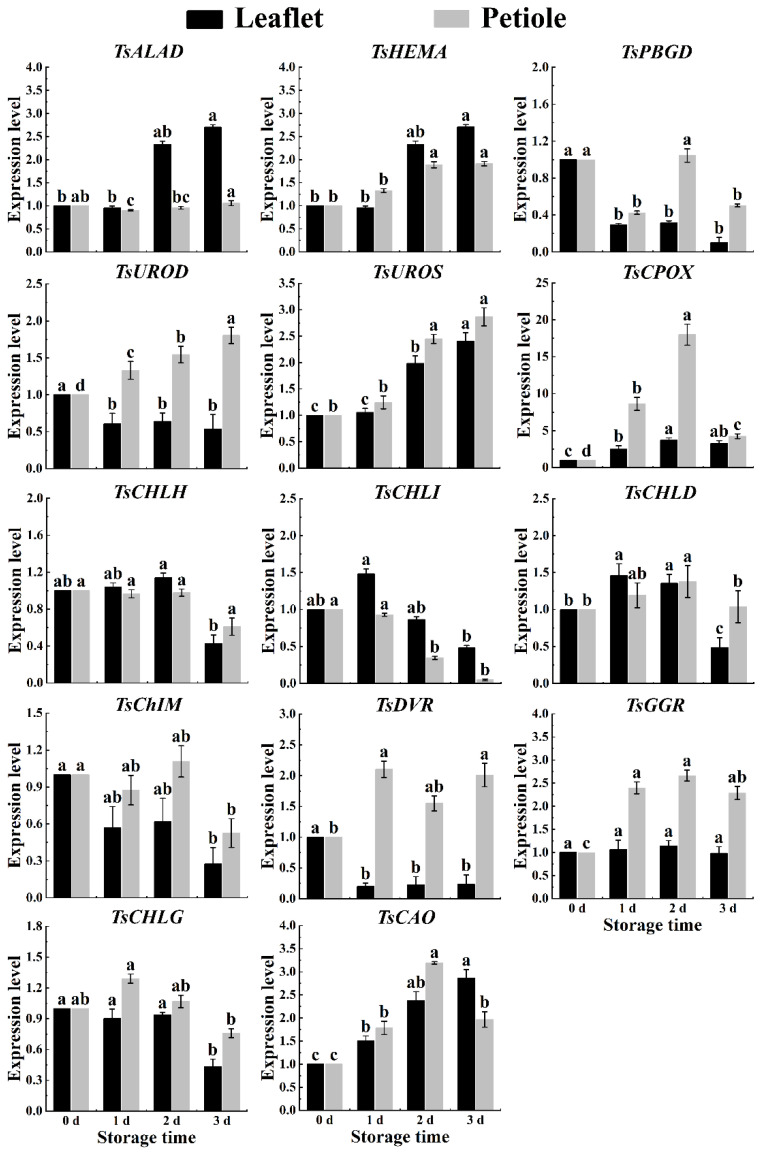
The chlorophyll-related gene expression level within the leaflet and petiole of postharvest spring red TSB through qRT-PCR, during four days of refrigerated storage (4 °C) in the dark. The “a”, “b”, “c”, and “d” indicate significant differences between treatments (*p* < 0.05).

**Figure 6 ijms-25-07719-f006:**
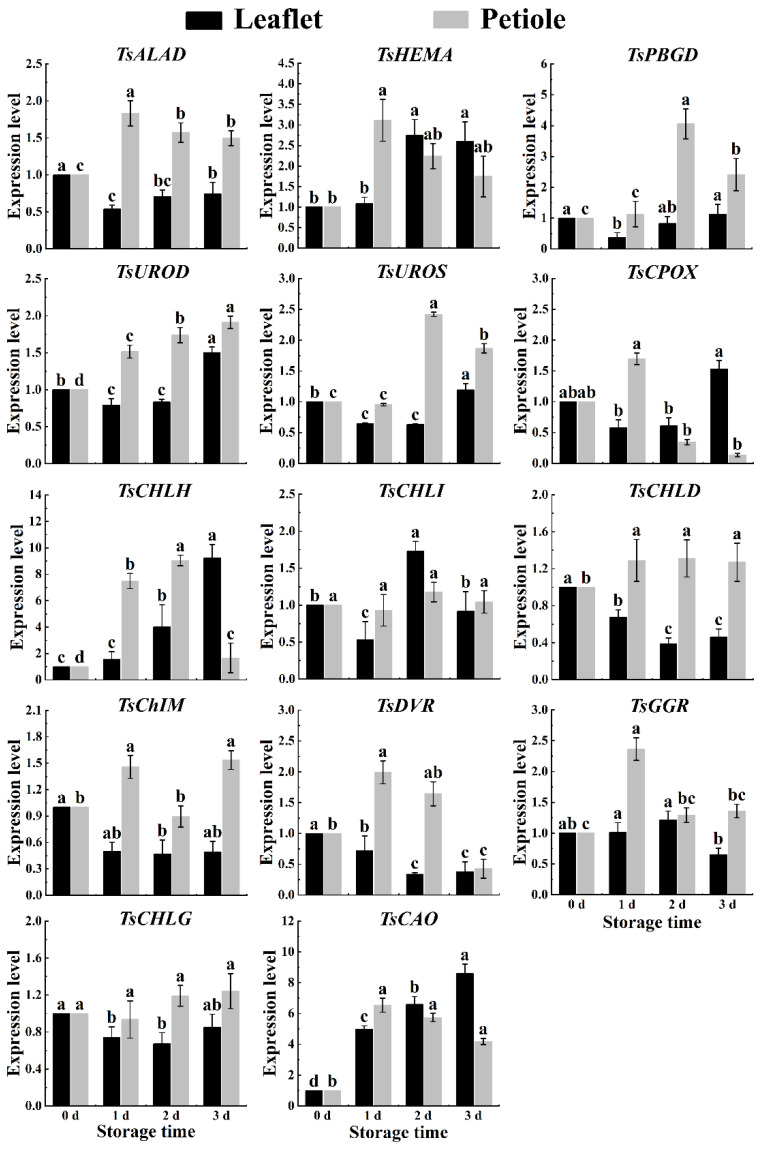
The chlorophyll-related gene expression level within the leaflet and petiole of postharvest summer red TSB through qRT-PCR, during four days of refrigerated storage (4 °C) in the dark. The “a”, “b”, “c”, and “d” indicate significant differences between treatments (*p* < 0.05).

**Figure 7 ijms-25-07719-f007:**
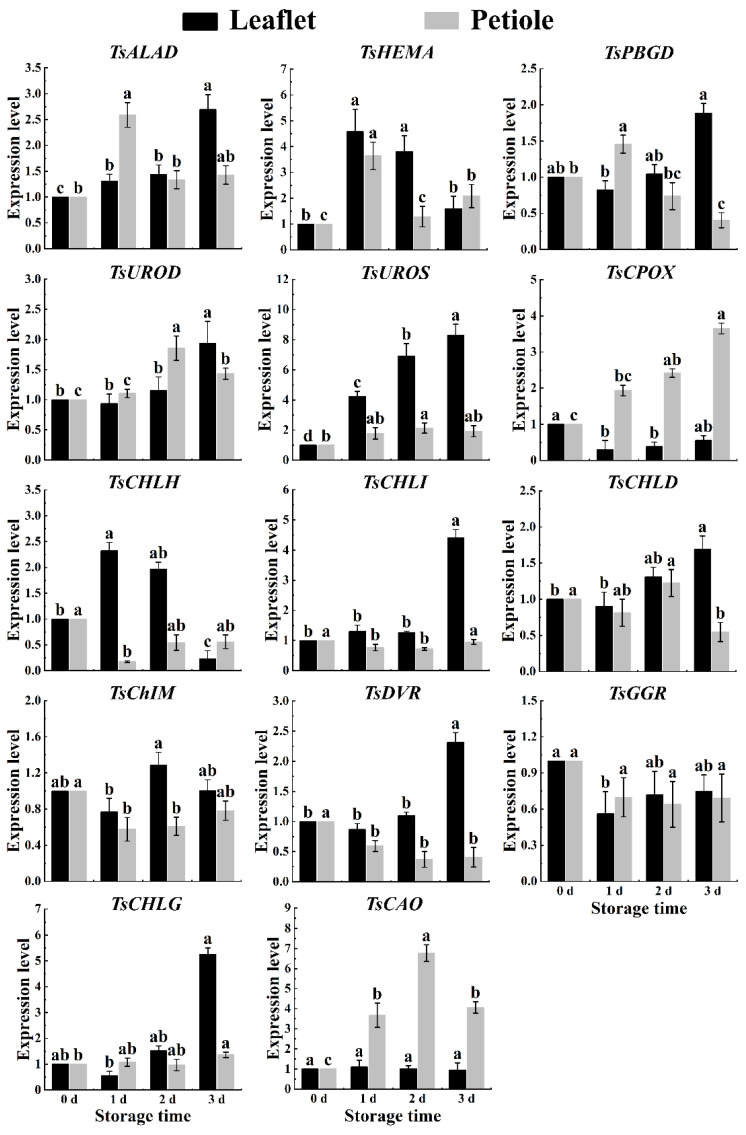
The chlorophyll-related gene expression level within the leaflet and petiole of postharvest autumn red TSB through qRT-PCR, during four days of refrigerated storage (4 °C) in the dark. The “a”, “b”, and “c” indicate significant differences between treatments (*p* < 0.05).

**Figure 8 ijms-25-07719-f008:**
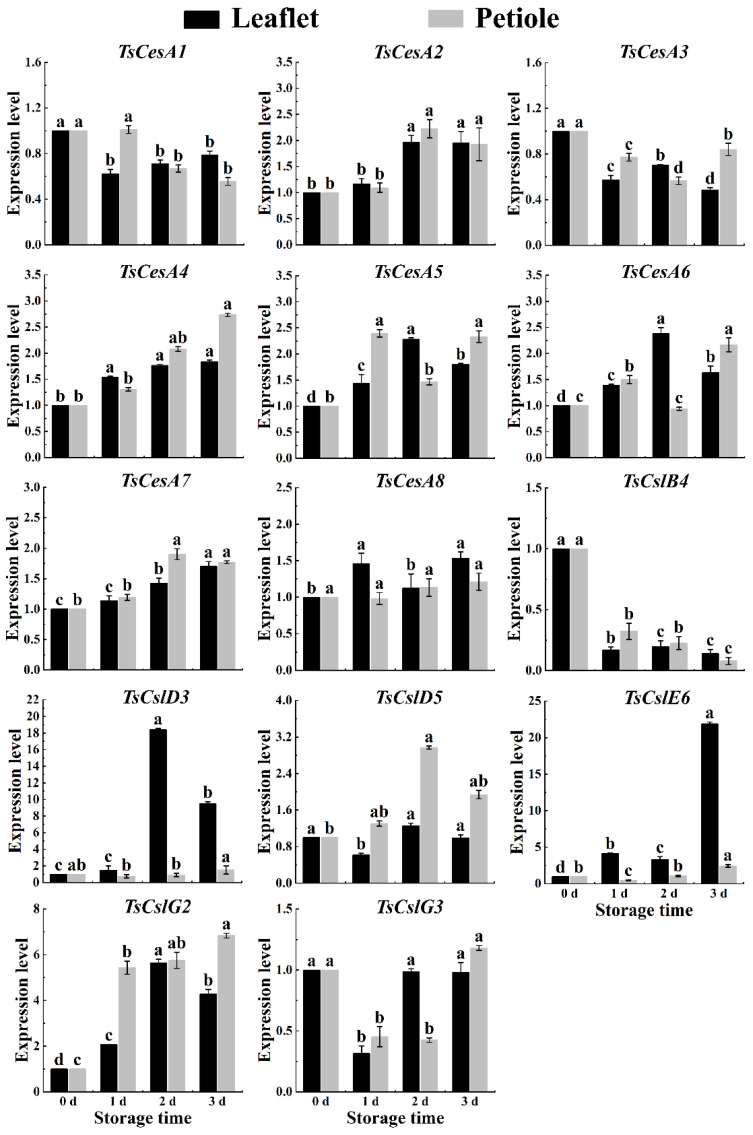
The cellulose-related gene expression level within the leaflet and petiole of postharvest spring red TSB through qRT-PCR, during four days of refrigerated storage (4 °C) in the dark. The “a”, “b”, “c”, and “d” indicate significant differences between treatments (*p* < 0.05).

**Figure 9 ijms-25-07719-f009:**
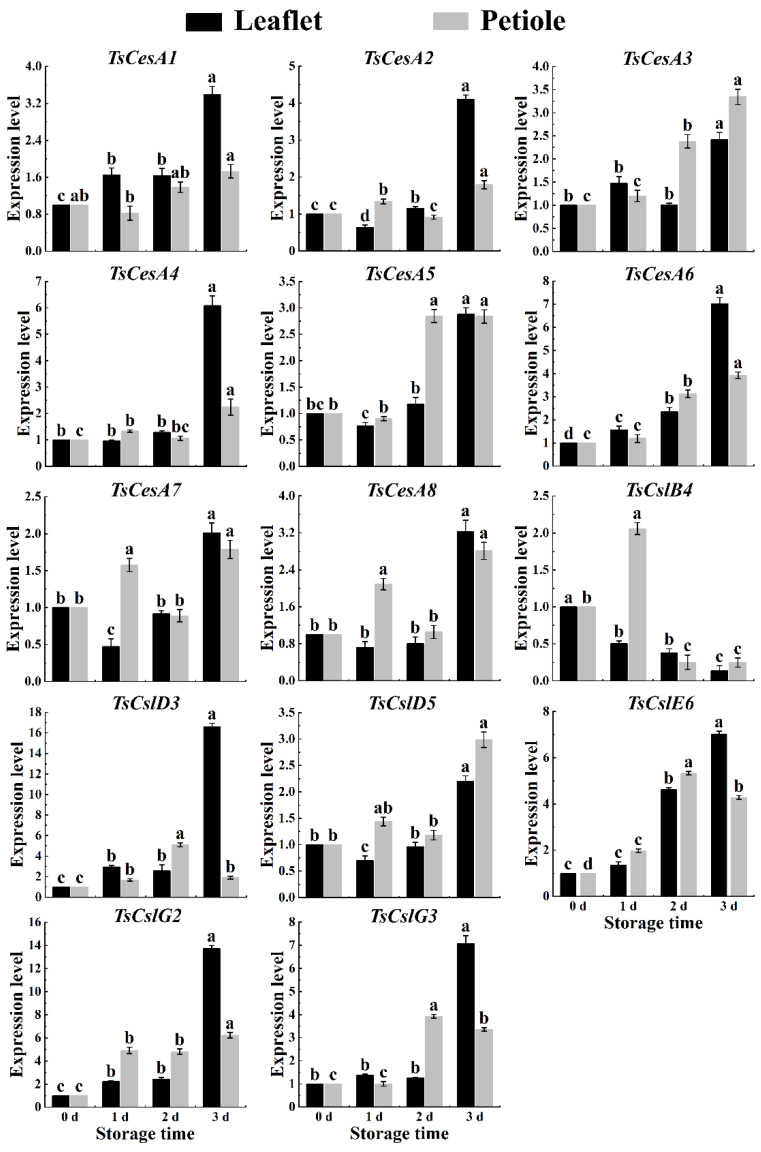
The cellulose-related gene expression level within the leaflet and petiole of postharvest summer red TSB through qRT-PCR, during four days of refrigerated storage (4 °C) in the dark. The “a”, “b”, “c”, and “d” indicate significant differences between treatments (*p* < 0.05).

**Figure 10 ijms-25-07719-f010:**
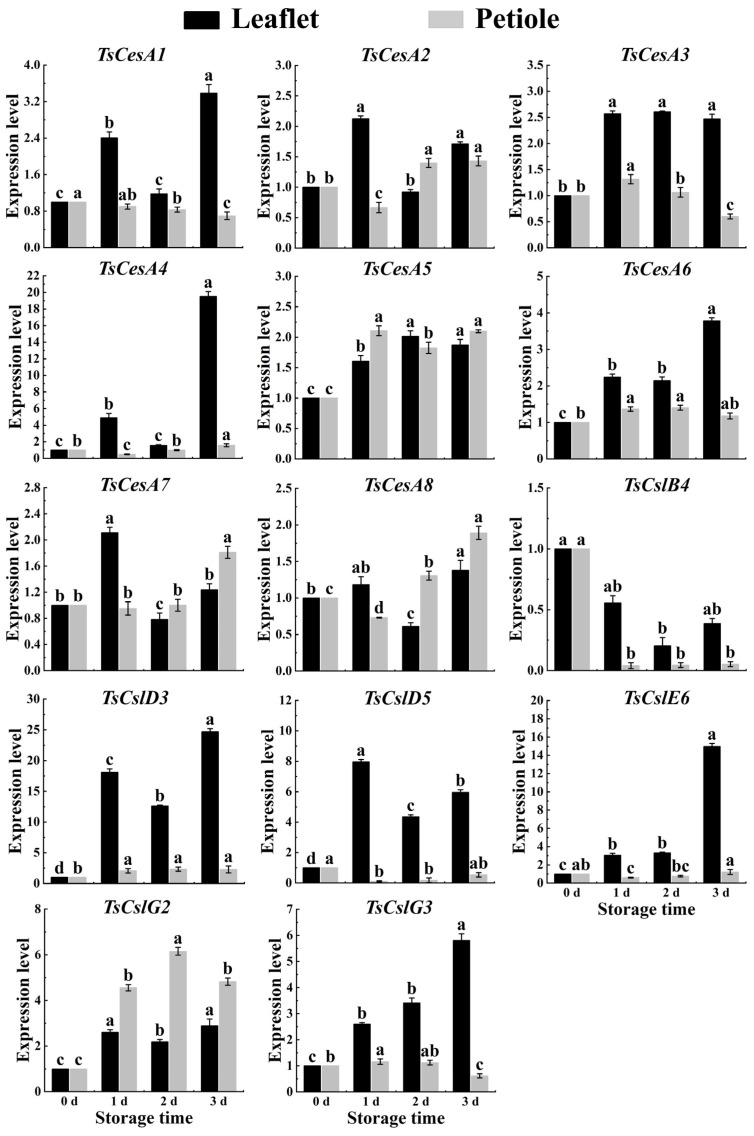
The cellulose-related gene expression level within the leaflet and petiole of postharvest autumn red TSB through qRT-PCR, during four days of refrigerated storage (4 °C) in the dark. The “a”, “b”, “c”, and “d” indicate significant differences between treatments (*p* < 0.05).

**Figure 11 ijms-25-07719-f011:**
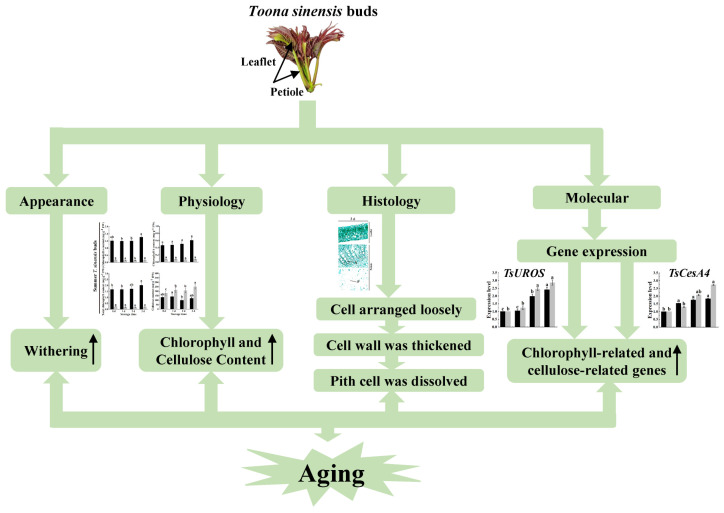
Diagram of TSB storage model during refrigeration.

## Data Availability

All data are reported in the article and Appendix A.

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
