# Peer review of "Low-Temperature Regulates the Cell Structure and Chlorophyll in Addition to Cellulose Metabolism of Postharvest Red Toona sinensis Buds across Different Seasons"

_ijms, 2024, doi:10.3390/ijms25147719_

Round 1
Reviewer 1 Report
Comments and Suggestions for Authors
In this ms, authors presented how chlorophyll and cellulose metabolism of postharvest RTSB is Regulated by cold temperature across seasons. Nicely presented but I have few concerns need to address such as:
1. All figures need to be self descriptive. I did not see the legends like that. Make it more comprehensible to the reader.
2. In Method, PCR analysis is too short. Please describe bit more. And, a brief of storage condition like aeration, light etc is needed
3. In result, 2.2 Ln 118, after 1 d, do you think cold was only factor? To me, moisture/water was another factor resulted here. explain this in the paragraph
4. In 136, What was the mechanism to loose the sponge tissue? Need to explain this
Others are fine. I am looking forward to see the revised ms with the information mentioned above.
3.
Reviewer 2 Report
Comments and Suggestions for Authors
This is a descriptive manuscript that intends to analyse the influence of cold storage in the dark upon leaf sprouts harvested at different seasons of the year.
Reading the manuscript, one gets the idea that there is a weakness on plant anatomy and basic knowledge on plant histology.
Minor comments.
Starting by the title: Note that there are not “Cold temperatures”. Temperatures can be low or high.
Figure 1 has excessive lettering. Since the figure is organized like a matrix, in the top of the columns the labels “Spring”, “Summer” and “Autumn” are sufficient instead of letters A, B and C in the pictures. The same applies to the numbers I, II, and III. In the beginning of each line the indication of the post-harvesting storage time (0 d, 1 d, 2 d and 3 d) is sufficient and clearer.
The same applies to figures 2 and 3. However, the pictures in this figure are too small and must be enlarged to allow detailed observation. As referred lettering (A – I and a- I to C - IV and c -IV) is excessive. “Spring”, “Summer” and “Autumn” in the top of the columns and “0 d”, “1 d”, “2 d” and “3 d” in the beginning of each line as appropriate are sufficient.
Considering that even after cutting and wounding the sprouts continue to develop, a comparative histological study with sprouts at the same developmental stage in planta must be included to analyse the effects of storage on the structure and histology of the harvested sprouts.
Regarding biochemical analyses and stating with the chlorophyll content, authors state as a discovery that “the chlorophyll a content was higher than that of chlorophyll b” without further explanation. It is known that plants have more chlorophyll a than b. In fact, most plant pigment is chlorophyll a. Chlorophyll a absorbs light better than chlorophyll b that is why there is more chlorophyll a in plants.
Regarding the results depicted on Figure 4, which intends to show ‘Chlorophyll and cellulose content during postharvest refrigeration of red T. sinensis sprouts from different seasons’, the graphs of Chlorophylls (a and b) could provide more information if they represent the chlorophylls content along storage time (0 - 3 days) for each harvest season.
Figures 5 and 6. The graphs are too small and do not allow for comfortable viewing. Only with a magnification of 200% was it possible to properly analyse the image. The graphs should be enlarged and organized into several larger images. Additionally and as previous referred for chlorophylls, the graphs of the relative gene expression, both for chlorophyll- and cellulose-related genes, could provide more information if they represent the gene expression levels along storage time (0 - 3 days) for each harvest season.
The Discussion is wordy, repeats results and does not fully explain the processes.
The “Methods and Materials” (usually is Material and Methods) have several inconsistencies, notably in the histologic technic or the missing formula of the chlorophylls determination, to refer a few examples.
By the way, and regarding the results. Authors expressed the results on a fresh weight basis. Considering that the sprouts lose water during storage, results should be expressed on a dry weight basis. This will show other results and may explain the inconsistencies.
But all of the previous comments are minor concerns.
The main criticism to this manuscript is related to terminology and incorrect concepts of histology and anatomy.
Firstly, it is important to understand that leaves, in general, are made up of a sheath, petiole and limb or blade. In other words, the petiole is part of the organ called the leaf. Thus, when authors refer to “leaf” and then “petiole” they should say “limb” and “petiole”.
Secondly, the leaves of Toona sinensis are compound leaves, that is, are leaves divided into leaflets or pinnules inserted into a rachis. Therefore, this compound leaf has petiole, rachis, petiolules and leaflets. So when they refer to “leaf” they should say the limb of a leaflet.
Once here, it is important to know the position of the leaflet that authors had analysed. This is because the internal histology of basal leaflets is different from the histology of apical leaflets during leaf development. That is, there is a gradation of histological differentiation in the leaflets from the base to the apex of the rachis during the initial phases of leaf development.
Then, when they refer to “petiole” it is important to know whether they are really referring to the petiole, the rachis or the petiolule.
Thirdly, it is important to ask what the authors mean by a “bud”. A Bud is an undeveloped shoot and typically occurs in the axil of a leaf or at the tip of a stem. Buds may be specialized to develop flowers or shoots. The buds of many woody plants are protected by modified leaves called scales, which tightly surround and protect the shoot apical meristem. When conditions are right bud opens and elongates and tiny leaves start to sprout.
Concluding, when authors refer to “buds” they should say leaf sprouting or sprouts.
Additionally, and in relation to leaf histology, authors must bear in mind that in several leaves the mesophyll is dorsiventrally organized into palisade parenchyma on the upper part and spongy parenchyma on the lower part. The vascular bundles (called veins) occupy the middle plane of the leaf and have the xylem on top and the phloem on the bottom.
That being said, there is no palisade tissue nor spongy tissue, as authors exhaustively refer to.
Scientific language must be rigorous and scientific terms have a precise meaning that is intended to be universal so that scientists can understand each other.
All the concepts mentioned above are basic anatomy and histology of leaves and buds that can be found in elementary plant anatomy texts (see for instance, Katherine Esau, 1953 - Plant Anatomy, Wiley).
Unfortunately, these errors, although published in MDPI journals, certainly escaped the reviewers. (Diversity 2022, 14, 572. https://doi.org/10.3390/d14070572, Agronomy 2023, 13, 119. https://doi.org/10.3390/agronomy13010119.
The above comments are sufficient enough, unfortunately, to not recommend the acceptance of this manuscript.
Comments on the Quality of English Language
English language and style need deep thorough revisions.
Round 2
Reviewer 1 Report
Comments and Suggestions for Authors
N/A
Author Response
Dear Editors and Reviewers,
Thank you very much for reviewing our manuscript ‘Low Temperature Regulates the Cell Structure and Chlorophyll and Cellulose Metabolism of Postharvest Red Toona sinensis Buds across Different Seasons’ (MANUSCRIPT ID: ijms-3038061). Your effort and time spent on our manuscript are greatly appreciated. We are delighted to all suggestion and review comments you made. Your critics/suggestions would definitely improve the quality of our manuscript.
This manuscript has been extensively edited according to your and reviewers’ comments. The manuscript was also critical read by one professional colleague in Canada whose native language is English. Please find the revised manuscript in International Journal of Molecular Sciences’s manuscript submission center.
We sincerely thank the editor and all reviewers for their valuable feed-back that helped us to improve the quality of our manuscript. The reviewers’ comments are laid out below in italicized font and specific concerns have been numbered. Our response is given in normal font and changes/editions to the manuscript is given in the red text.
Thank you again for your kind consideration and excellent suggestions toward our manuscript. We hope these revisions would be sufficient to satisfy ijms publication consideration. We are looking forward to hearing from you soon.
Yours sincerely
Reviewer 2 Report
Comments and Suggestions for Authors
The authors have improved the English and some suggestions have been taken into consideration. However, English still needs a thorough revision and important amendments were not seriously considered, namely terminology and concepts of plant anatomy and plant histology.
Starting with the terminology. What is known is that the young leaves from Toona sinensis trees are widely used as a vegetable in Chinese cuisine, with the young red leaves being preferred for the best flavour.
This referee has already written what follows bellow but was probably not understood. I am going to explain in other words. The Toona sinensis leaf is a good example of a pinnately compound leaf, it has a lamina (= blade) divided into separate leaflets (= pinnula). The petiole extends from the point of connection in the branch to the first leaflet. From this first leaflet to the tip, the central axis is called rachis where other leaflets are inserted through the petiolules.
Along the text, authors changed the word petiole to shoot. However, they are not the same thing. A plant shoot is the above-ground part of a plant. It may be considered, as well, any stem together with branches and its appendages like leaves and buds (apical, lateral and flowering buds).
Analysing previous works published by the first author, the petiole might be the only structure that was correctly referred to in the first version of this manuscript.
Authors insist to call “buds” to the sets of tender compound-leaves they analysed. The young compound-leaves sprouting from a small branch are not a bud. A bud, as previously referred in the first revision, is a meristematic structure (a shoot apical meristem) surrounded and protected by scales. A bud normally occurs in the axil of a leaf or at the tip of a stem. When conditions are right the bud burst, opens and elongates and, in the case of Toona sinensis plant, tiny compound leaves start to develop.
Therefore, when authors refer to “buds” (meaning the set of young developing leaves, which they harvested and analysed) they should say “leaf sprouts”, “young leaves” or, taking into account the second definition of shoot, “young leafy-shoots” (meaning the young branch developed from a bud plus the tender compound leaves developing from leaf primordia on the shoot apical meristem).
As referred in the first round of revision, there are neither palisade tissue nor spongy tissue. The tissue is uniquely parenchyma that is organized in palisade parenchyma and spongy parenchyma. This parenchyma is often referred to as chlorenchyma because parenchyma cells are filled with chloroplasts. These concepts must be corrected throughout the text.
Considering that the red leaf sprouts (or red young leaves/red young-leafy-shoots) lose water during storage for the four days analysed, all biochemical determinations expressed in a fresh mass basis must take this loss of water into account.
The authors reported an increase in chlorophyll content during storage, this can occur, but if there is a water loss over time, the chlorophyll content expressed in a fresh mass basis, will obviously increase. As previously recommended results should be expressed on a dry mass basis.
Authors argue that the increase in chlorophyll and cellulose contents are correlated with the increase in the expression of more than half of the chlorophyll-related and cellulose-related genes during refrigerated storage. However, the increased expression of these transcripts may be more related with the processes that would occur during normal development, rather than with the storage. Processes like the future synthesis of proteins/enzymes for photosynthetic systems and for cell wall synthesis (which is needed to accompany cell elongation and new cells formation).
As for the thickening of the secondary cell wall of vessel elements reported by the authors to occur during storage, this process is related with the normal differentiation of xylem conducting elements.
The lysis of medullary cells, cell shrinkage and concomitant increase of intercellular spaces it is likely to be related with the water loss. To distinguish between autolysis of pith cells, which can occur in a few stems, and lysis resulting from cutting injury, it is necessary to analyse the same structure in the plant.
As previously referred, after cutting the sprouts/young leafy-shoots continue to develop. Therefore, a comparative histological study with young leafy-shoots (sprouts) at the same developmental stage in planta, must be included to allow a meaningful analyse of the storage effects on the structure and histology of the harvested young leafy-shoots (sprouts).
Figure 4. In order to make a meaningful comparison of the effect of cold storage on sprouts (young leafy-shoots) harvested in each season, the contents of both chlorophyll and cellulose must be represented along the 4 days of storage in each season. That is, chlorophyll content on days 0, 1, 2 and 3 of the sprouts harvested in the Spring; chlorophyll on days 0, 1, 2 and 3 of the sprouts harvested in the Summer and chlorophyll on days 0, 1, 2 and 3 of the sprouts harvested in the Autumn. The same for the cellulose content.
Graphs are still too small. Units must indicate whether they are expressed by fresh weight or dry weight. For chlorophylls preferably dry-height in a new version of the manuscript.
Similarly, and as referred in the first revision round, the graphs (Figures 5 and 6) of the relative gene expression, both for chlorophyll synthesis- and cellulose synthesis-related genes, will provide more information if they represent the gene expression levels along storage time (0 - 3 days) for each harvest season. Nonetheless, the graphs are still very small, they need to be enlarged to 200% to be viewed correctly.
By the way, in the y-axis label correct "expressive level" to expression level.
Some suggestions for better explanation of captions and text follow.
Line 71 – “The initial buds of TSB are often red …” even if this referee could accept the term bud, note the redundancy in the sentence where B, in the abbreviation TSB, stand for bud(s). This would be “The initial buds of Toona sinensis buds are often red ...” Change to “The initial leafy-shoots of Toona sinensis are often red …” or similar.
Line 74 – correct to “…and the entire young shoot, tender leaflets and petioles can be eaten…” or similar.
Line 111. “Therefore, red TSB planted in Guizhou province was used as the experimental material…”. TSB stands for Toona sinensis buds. Buds have not been planted!
Caption to Figure 1. “Colour change of the red young-leaves of Toona sinensis harvested at different times of the year, during four days of refrigerated storage (4 °C) in the dark”.
Line 152 – “Palisade tissue and spongy tissue are an important part of the chloroplasts … !” Chloroplasts are organelles that occur abundantly in cells of photosynthetic tissues, such as the parenchyma in the leaf mesophyll. The sentence must be re-written.
Caption to Figure 2. “Mesophyll structure of the leaflets from red young-shoots harvested at different times of the year, during 4 days of refrigerated storage (4 °C) in the dark”.
Caption to Figure 3. “Petiole structure of young red compound-leaves from young shoots harvested in different seasons, and stored for four days under refrigeration (4 °C) in the dark”. Explain/describe A and B.
In the graphs of chlorophyll and cellulose contents, the results are expressed in mg.g-1 (more understandable units); however, the text still presents the old values expressed in g.kg-1.
Lines 353 -. “The different seasons of red TSB exhibited wilting and a loss of water at one day of postharvest storage”. Seasons do not exhibit wilting and a loss of water
Lines 369 -. “… which also means that the storage time of the spring and autumn red TSB is shorter than that of the summer red TSB”. The storage time was the same (4 days). The shelf life can be different.
Lines 374- 376. What do authors intend to substantiate with “This finding was consistent with the phenomenon of the formation of voids by autolysis of the cells during the occurrence of the pineapple fruit pulp hydatid disease”? (A reference is needed to understand the meaning of this phrase).
Lines 437 – 440. Authors say that “This is at odds with the prevailing trend of gene expression levels related to chlorophyll metabolism during postharvest cold storage of toon buds, which may be attributable to plant specific factors.” As this referee has already explained, the discrepancy comes from the loss of water that the authors did not take into account when determining the chlorophyll content.
Line 487 - The authors report that they determined the OD at wavelengths of 665 nm, 649 nm and 470 nm. Why reading at 470 nm if the carotenoids have not been determined?
The formula of the chlorophylls determination is out of place and should be inserted at the end of the previous paragraph.
Standardize the spelling of temperature in degrees Celsius (°C). Place a space between the number and the °C symbol. Example: 4 °C (and not 4°C), -80 °C (and not - 80 º C), 95 °C (and not 95 ° C).
Numbering of references need to be carefully checked.
For instance, reference 49 is referred for Chlorophyll determination method. However, in the respective paper we cannot find any chlorophyll determination: “49. Liu, J.X.; Feng, K.; Wang, G.L.; Xu, Z.S.; Wang, F.; Xiong, A.S. Elevated CO2 induces alteration in lignin accumulation in 682 celery (Apium graveolens L.). Plant Physio. Biochem. 2018, 127, 310-319”.
All previous comments and suggestions are small examples of the lack of care in the manuscript preparation. They also illustrate that the experimental work was not well planned and presents analytical errors.
Therefore, the manuscript cannot be considered for publication.
Comments on the Quality of English Language
An English-speaking author revised the manuscript; however, this referee considers that a thorough revision is still necessary. Along the text, the concordance between subject and verb must be checked. Several sentences have the subject in the wrong position are meaningless.
Author Response
Dear Editors and Reviewers,
Thank you very much for reviewing our manuscript ‘Low Temperature Regulates the Cell Structure and Chlorophyll and Cellulose Metabolism of Postharvest Red Toona sinensis Buds across Different Seasons’ (MANUSCRIPT ID: ijms-3038061). Your effort and time spent on our manuscript are greatly appreciated. We are delighted to all suggestion and review comments you made. Your critics/suggestions would definitely improve the quality of our manuscript.
This manuscript has been extensively edited according to your and reviewers’ comments. The manuscript was also critical read by one professional colleague in Canada whose native language is English. Please find the revised manuscript in International Journal of Molecular Sciences’s manuscript submission center.
We sincerely thank the editor and all reviewers for their valuable feed-back that helped us to improve the quality of our manuscript. The reviewers’ comments are laid out below in italicized font and specific concerns have been numbered. Our response is given in normal font and changes/editions to the manuscript is given in the red text in the word file.
Thank you again for your kind consideration and excellent suggestions toward our manuscript. We hope these revisions would be sufficient to satisfy ijms publication consideration. We are looking forward to hearing from you soon.
Yours sincerely
